# DexDeform: Dexterous Deformable Object Manipulation with Human Demonstrations and Differentiable Physics

**Sizhe Li[1],[∗], Zhiao Huang[2],[∗], Tao Chen[1], Tao Du[3],[4],**
**Hao Su[2], Joshua B. Tenenbaum[5], Chuang Gan[6],[7]**
[1]MIT, [2]UC San Diego, [3]Tsinghua University, [4]Shanghai Qi Zhi Institute,
[5]MIT BCS, CBMM, CSAIL, [6]UMass Amherst, [7]MIT-IBM Watson AI Lab
`sizheli@csail.mit.edu, z2huang@eng.ucsd.edu, taochen@mit.edu,`
`taodu@tsinghua.edu.cn, haosu@eng.ucsd.edu, jbt@mit.edu,`
`chuangg@umass.edu`

## Abstract

In this work, we aim to learn dexterous manipulation of deformable objects using multi-fingered hands. Reinforcement learning approaches for dexterous rigid object manipulation would struggle in this setting due to the complexity of physics interaction with deformable objects. At the same time, previous trajectory optimization approaches with differentiable physics for deformable manipulation would suffer from local optima caused by the explosion of contact modes from hand-object interactions. To address these challenges, we propose DexDeform, a principled framework that abstracts dexterous manipulation skills from human demonstration, and refines the learned skills with differentiable physics. Concretely, we first collect a small set of human demonstrations using teleoperation. And we then train a skill model using demonstrations for planning over action abstractions in imagination. To explore the goal space, we further apply augmentations to the existing deformable shapes in demonstrations and use a gradient optimizer to refine the actions planned by the skill model. Finally, we adopt the refined trajectories as new demonstrations for finetuning the skill model. To evaluate the effectiveness of our approach, we introduce a suite of six challenging dexterous deformable object manipulation tasks. Compared with baselines, DexDeform is able to better explore and generalize across novel goals unseen in the initial human demonstrations. Additional materials can be found at our project website [1].

## 1 Introduction

The recent success of learning-based approaches for dexterous manipulation has been widely observed on tasks with rigid objects (OpenAI et al., 2020; Chen et al., 2022; Nagabandi et al., 2020). However, a substantial portion of human dexterous manipulation skills comes from interactions with deformable objects (e.g., making bread, stuffing dumplings, and using sponges). Consider the three simplified variants of such interactions shown in Figure 1. **Folding** in row 1 requires the cooperation of the front four fingers of a downward-facing hand to carefully lift and fold the dough. **Bun** in row 4 requires two hands to simultaneously pinch and push the wrapper. Row 3 shows **Flip**, an in-hand manipulation task that requires the fingers to flip the dough into the air and deform it with agility.

In this paper, we consider the problem of deformable object manipulation with a simulated Shadow Dexterous hand (ShadowRobot, 2013). The benefits of human-level dexterity can be seen through the lens of versatility (Feix et al., 2015; Chen et al., 2022). When holding fingers together, the robot hands can function as a spatula to fold deformable objects (Fig. 1, row 1). When pinching with fingertips, we can arrive at a stable grip on the object while manipulating the shape of the object (Fig. 1, row 2). Using a spherical grasp, the robot hands are able to quickly squeeze the dough into a

---

[∗]Equal Contribution

[1]Project website: `https://sites.google.com/view/dexdeform`

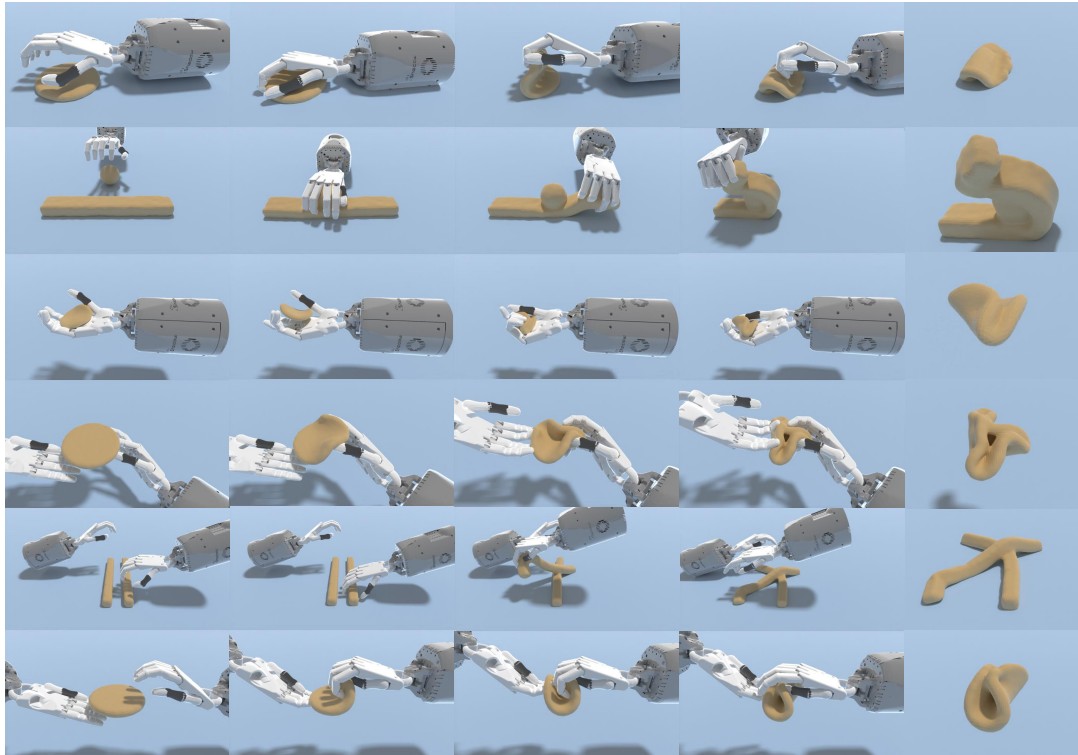

Figure 1: We present a framework for learning dexterous manipulation of deformable objects, covering tasks with a single hand (**Folding** and **Wrap**, row 1-2), in-hand manipulation (**Flip**, row 3), and dual hands (**Bun**, **Rope**, **Dumpling**, row 4-6). Images in the rightmost column represent goals.

folded shape (Fig. 1, row 3). Therefore, it is necessary and critical to learn a manipulation policy that autonomously controls the robot hand with human-like dexterity, with the potential for adapting to various scenarios. Additionally, using a multi-fingered hand adds convenience to demonstration collection: (1) controlling deformable objects with hands is a natural choice for humans, resulting in an easy-to-adapt teleoperation pipeline. (2) there exists a vast amount of in-the-wild human videos for dexterous deformable object manipulation (e.g., building a sand castle, making bread). Vision-based teleoperation techniques can be employed for collecting demonstrations at scale (Sivakumar et al., 2022). As with any dexterous manipulation task, the contact modes associated with such tasks are naturally complex. With the inclusion of soft bodies, additional difficulties arise with the tremendous growth in the dimension of the state space. Compared to the rigid-body counterparts, soft body dynamics carries infinite degrees of freedom (DoFs). Therefore, it remains challenging to reason over the complex transitions in the contact state between the fingers and the objects.

Given the high dimensionality of the state space, the learning manipulation policy typically requires a large number of samples. With no or an insufficient amount of demonstrations, interactions with the environment are needed to improve the policy. Indeed, past works in dexterous manipulation have leveraged reinforcement learning (RL) approaches for this purpose (Rajeswaran et al., 2017; Chen et al., 2022). However, the sample complexity of most RL algorithms becomes a limitation under the deformable object manipulation scenarios due to the large state space. Recent works have found trajectory optimization with the first-order gradient from a differentiable simulator to be an alternative solution for soft body manipulation (Huang et al., 2021; Li et al., 2022a; Lin et al., 2022). However, the gradient-based optimizers are found to be sensitive to the initial conditions, such as contact points. It remains unclear how to leverage the efficiency of the gradient-based optimizer and overcome its sensitivity to initial conditions at the same time.

In this work, we aim to learn dexterous manipulation of deformable objects using multi-fingered hands. To address the inherent challenges posed by the high dimensional state space, we propose DexDeform, a principled framework that abstracts dexterous manipulation skills from human demonstrations and refines the learned skills with differentiable physics. DexDeform consists of three components:

(1) collecting a small number of human demonstrations (10 per task variant) with teleoperation for initializing the training data. (2) extracting abstractions of the dexterous action sequences from demonstrations with a skill model. This model decomposes the manipulation process and allows for planning for a novel goal with a learned skill dynamics predictor. (3) using differentiable physics to refine trajectories planned by the skill model on augmented goals, which adds new trajectories to further fine-tune the skill model. Hence, DexDeform is capable of avoiding local minima of the gradient-based optimizer by initializing trajectories with the abstractions of dexterous actions. At the same time, DexDeform enjoys the efficiency of the gradient-based optimizer to augment demonstrations for bootstrapping the learned skill model.

To evaluate the effectiveness of DexDeform, we propose a suite of six challenging dexterous deformable object manipulation tasks with a differentiable simulator. Extensive experiment results suggest that DexDeform can successfully accomplish the proposed tasks and explore different goal shapes on a set of dexterous deformable object manipulation tasks. In summary, our work makes the following contributions:

- We perform, to the best of our knowledge, the first investigation on the learning-based dexterous manipulation of deformable objects.
- We build a platform that integrates a low-cost teleoperation system with a soft-body simulation that is differentiable, allowing humans to provide demonstration data.
- We propose DexDeform, a principled framework that abstracts dexterous manipulation skills from human demonstration, and refines the learned skills with differentiable physics.
- Our approach outperforms the baselines and successfully accomplishes six challenging tasks such as **Flip**, learning complex soft-body manipulation skills from demonstrations.

## 2 METHOD

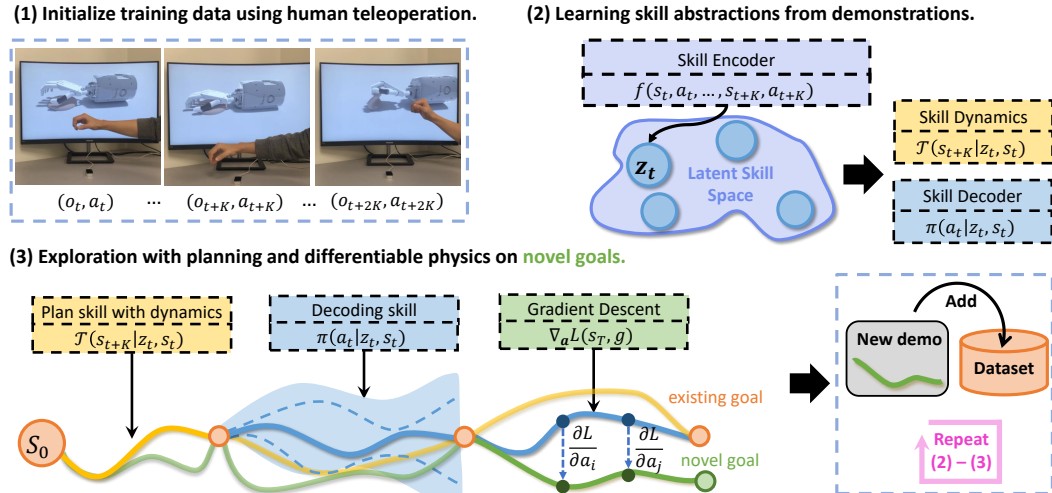

Figure 2: **An overview of DexDeform** (Sec.2). (1): We first collect human demonstrations using hand tracking teleoperation. (2): We then train a skill model from demonstrations, which consists of a skill sequence encoder, a skill dynamics predictor, and a skill action decoder. (3): To explore the high dimensional state space, we use the skill model to plan for novel goals, and apply a gradient-based optimizer to refine the actions planned by the skill model. Lastly, we store the successful trajectories as new demonstrations and repeat (2)-(3).

Given a goal configuration of deformable shapes, our objective is to plan actions to perform dexterous deformable object manipulation using the Shadow Dexterous hand. We assume we know the full point cloud observation of the scene, which includes the multi-fingered hand(s) and the object(s).

To tackle this problem, we propose DexDeform (Fig. 2), a framework that consists of three components: (1) a small set of human demonstrations for policy initialization (Sec. 2.1); (2) learning

skill abstractions of dexterous actions from demonstrations (Sec. 2.2); (3) exploring novel goals with planning and differentiable physics (Sec. 2.3).

## 2.1 COLLECTION OF HUMAN MANIPULATION DEMONSTRATION

Dexterous manipulation with a high degrees-of-freedom robot hand poses challenges for policy learning, since the state dimension is high and dexterous tasks involve frequent contact making and breaking between the hand and the objects. Learning such a policy from scratch would be extremely time consuming (OpenAI et al., 2020). One way to overcome the exploration challenge is by providing human demonstration data. However, many prior works use a complex and expensive system such as a motion capture system (Rajeswaran et al., 2017; Gupta et al., 2016) or many cameras (Handa et al., 2020) for capturing human demonstration data. We built a low-cost ($100) and simple teleoperation system that allows a human operator to control the simulated hand in real time to perform dexterous manipulation tasks. Our system is built based on the Leap Motion Controller (Spiegelmock, 2013) which is an optical hand tracking device. By constructing an inverse kinematics model based on the Shadow hand, our system re-targets the detected human finger positions into the joint positions of the simulated robot hand that is controlled via a position-based PD controller. More details on teleoperation setup can be found in Appendix B.

## 2.2 LEARNING ABSTRACTIONS OF DEXTEROUS SKILLS

Humans execute abstractions of dexterous skills to interact with deformable objects instead of planning every finger muscle movement involved. In the same spirit, we would like to learn abstractions of actions present in the collected human demonstrations. Our skill model consists of three components, a skill encoder, a skill dynamics predictor, and a skill action decoder. The skill model uses dynamics predictor for planning, and action decoder to predict actions from skill embeddings. The skill model is built on the implicit scene representations of point clouds, which we will describe first.

**Implicit Representation of the Scene.** We leverage the translational equivariance offered by the Convolutional Occupancy Network (Peng et al., 2020), or ConvONet, to build a continuous implicit representation of the scene. Concretely, let $o_t \in \mathcal{O}$ describe the unordered point cloud observation at time $t$, where $o_t = \{x_1, x_2, ..., x_n\}$ with $x_i \in \mathbb{R}^6$ (the 3D position and 3D color label). The encoder of ConvONet $\psi_{enc} : \mathcal{O} \to \mathbb{R}^{H \times W \times D}$ maps a point cloud to a set of 2D feature maps. Given a query point $p \in \mathbb{R}^3$, we get its point feature $\psi_{enc}(o_t)|_p$ from the feature maps $\psi_{enc}(o_t)$ via bilinear interpolation. An occupancy decoder $\psi_{dec}(p, \psi_{enc}(o_t)|_p) : \mathbb{R}^3 \times \mathbb{R}^D \to \mathbb{R}^3$ is then used to map a query point $p$ and its point feature $\psi_{enc}(o_t)|_p$ into the occupancy probabilities of being free space, hand, and the deformable object, based on one-hot encoding. Our ConvONet is trained with self-supervision in simulation. We use the 2D feature maps from the encoder as the translational-equivariant representation of the scene.

**Latent encoding of the implicit scene representation.** As will be explained later, our choice of implicit representation of the scene can be naturally integrated with our skill model for planning for target deformable shapes. To extract a compact scene representation for dynamics modeling and planning in the latent space, we train a VAE to reconstruct the 2D feature maps from ConvONet encoder outputs. The VAE includes an encoder $\phi_{enc}$ that encodes the scene representation into a latent vector $s_t$ and a decoder $\phi_{dec}$ that decodes the latent back into the scene representation space. Specifically, $s_t = \phi_{enc}(\psi_{enc}(o_t))$, and $\phi_{dec}(s_t) := \psi_{enc}(o_t)$.

**Skill Encoder.** Using our learned latent encoding of the scene, we encode the point cloud observation $o_t$ at each timestep into $s_t$. We then train a skill encoder $f$ that maps a sequence of $K$-step observation-action pairs $(s_t, a_t, ..., s_{t+K}, a_{t+K})$ into a skill embedding space containing $z_t$, i.e., $z_t \sim f(s_t, a_t, ..., s_{t+K}, a_{t+K})$. We use $z_t$ for decoding actions between timesteps $t$ and $t + K$.

**Skill Dynamics and Skill Decoder.** For each skill embedding, we follow SkiMo (Shi et al., 2022b) to jointly learn to predict the resulting dynamics of applying the skill and decode the actions responsible for forming the skill. Concretely, we train a skill dynamics predictor $\hat{s}_{t+K} = \mathcal{T}(z_t, s_t)$ that predicts the future latent scene encoding $K$ steps away in imagination. We also train a skill action decoder $\pi(a_t|z_t, s_t)$ that recovers the actions corresponding to the skill abstractions. We refer the readers to Appendix C for the training details and objective functions.

**Long-horizon planning in the space of skill abstractions.** Our choice of implicit representation of the scene allows us to decode the latent encoding for computing the shape occupancy loss for planning. Given a target shape $\mathbf{g}$ described by a point cloud and horizon $H$, we hope to find the sequence of skills $z_1, z_K, ..., z_H$ such that the final predicted scene encoding $\hat{s}_{H+K}$ is occupied by the points in target shape under the object category. Let $\mathcal{L}_{occ}(\mathbf{g}, h_{H+K})$ be the sum of the cross entropy loss computed between each point $x_i$ within target shape $\mathbf{g}$ described as a point cloud and the predicted occupancy probabilities $\psi_{dec}(x_i, \phi_{dec}(s_{H+K}))$ when $x_i$ is queried in the decoded scene representation. We formulate our planning problem as a continuous optimization problem.

$$\underset{z_1, z_K, ..., z_H}{\arg\min} \; C(\mathbf{g}, \mathbf{z}) = \mathcal{L}_{occ}(\mathbf{g}, \mathcal{T}(z_H, \mathcal{T}(z_{H-K}, ...\mathcal{T}(z_1, s_1)))) \qquad (1)$$

Here, $\mathbf{z} = z_1, z_K, ..., z_H$ is the sequence of skills we are optimizing over, and we iteratively apply $\mathcal{T}$ in a forward manner $\lfloor H/K \rfloor$ times to predict the resulting scene encoding $\hat{s}_{H+K}$. In practice, we optimize a batch of initial solutions $\{z_1, z_K, ..., z_H\}_{j=1}^J$ and choose the best one based on $C(\mathbf{g}, \mathbf{z})$. We refer the readers to Appendix C for more details on skill planning.

## 2.3 DIFFERENTIABLE PHYSICS GUIDED EXPLORATION

Given that the skill model could be limited to interactions captured in the current demonstration set, more interactions are needed for the skill model to generalize across novel goals. Two challenges exist: (1) Given the high degrees of freedom of soft bodies and human demonstrations, our shape distribution cannot be easily defined in closed-form expressions. How can we sample novel goals in the first place? (2) Suppose that a novel target shape is provided and is not closely captured by the demonstration set the skill model is trained on. How can we efficiently enable the skill model to achieve the novel shape, which would allow us to expand our demonstration set? We present two ideas for overcoming the two challenges.

**Shape augmentation for novel goal generation.** To tackle the intractability of the distribution of deformable shapes and sample new shapes, we explore the space of deformable shapes based on the shapes covered by the current demonstrations. We employ two simple geometric transformations: translation on the xz-plane and rotation around the y-axis, which is similar to data augmentation practices in training neural networks for image classification. We randomly sample target shapes from the existing demonstrations and apply augmentations to generate new target shapes.

**Differentiable-physics based trajectory refinement.** We use trajectories planned by the skill model as optimization initialization to overcome the local optima caused by the complex contacts, and use the gradient-based optimizer to refine the planned trajectories within tens of iterations.

**The DexDeform algorithm.** Putting all ingredients together, we present the DexDeform algorithm (Algo. 2) in Appendix D. During training, our framework learns implicit scene representations and the skill model. During exploration, our framework leverages a differentiable physics optimizer to expand the demonstrations to new goals.

## 3 EXPERIMENTS

In this section, we conduct experiments aiming to answer the following questions:

- Q1: How does DexDeform compare against trajectory optimization, imitation learning, and RL approaches?
- Q2: How much improvement does differentiable physics guided exploration bring for solving dexterous deformable manipulation tasks?
- Q3: What are the benefits of the skill model?
- Q4: Are skill dynamics predictions consistent with the resulting states from applying the decoded actions?
- Q5: What does the latent space of skill embedding look like?

## 3.1 ENVIRONMENTAL SETUP

**Tasks and Environments.** Inspired by human dexterous deformable object manipulation tasks, we design six tasks (Fig. 1): three single-hand tasks (**Folding**, **Wrap**, **Flip**), including in-hand

manipulation, and three dual-hand tasks (**Rope**, **Dumpling**, **Bun**). Detailed descriptions of our environments and tasks can be found in Appendix A. Each Shadow hand has 28 degrees of freedom with a movable base.

**Human Demonstration Collection.** Using our teleoperation setup described in Section 2.1, We collected 10 demonstrations for each task variant. There exist 4 variants for **Folding**, corresponding to left, right, front, and back folding directions. All other tasks have 1 task variant. The total demonstration amounts to approximately $60,000$ environment steps, or 2 hours of human interactions with the environment.

**Evaluation metric.** We report the normalized improvement (i.e., decrease) in Earth Mover distance (EMD) computed as $d(t) = \frac{d_0 - d_t}{d_0}$, where $d_0$ and $d_t$ are the initial and current values of EMD. Hence, a normalized improvement score of 0 represents a policy that results in no decrease in the EMD, while a score of 1 indicates that the policy is able to result in a shape that matches perfectly with the goal. We threshold the minimum of the score to be 0, as negative distances could occur if the policy results in shapes further away from the goal shape than the initial one. We approximate the EMD using Sinkhorn Divergence (Séjourné et al., 2019) between the source and target particles to quantify the fine-grained difference between a state and a goal.

**Baselines.** We consider four categories of baselines:
- **Model-free Reinforcement Learning.** We compare against Proximal Policy Optimization (PPO) (Schulman et al., 2017), an model-free RL method. The RL agent takes point cloud as input.
- **Behavior Cloning.** We compare with a baseline that directly learns a goal-conditioned policy with Behavior Cloning (BC) and hindsight relabeling. The agent is trained with the same human demonstration set and takes point cloud as input.
- **Model-free Reinforcement Learning with Data Augmentation.** We compare with demonstration augmented policy gradient (DAPG), a method that combines demonstrations with an RL agent (PPO). We train the agent using the same demonstration set with point cloud as input observations.
- **Trajectory Optimization.** We compare against the trajectory optimization (TrajOpt) that uses first-order gradients from a differentiable simulator to solve for an open-loop action sequence (Kelley, 1960). This method takes the full state of the simulation as the observation at each timestep.

## 3.2 Object Manipulation Results

Given a goal configuration of deformable shapes, our objective is to perform dexterous deformable object manipulation using the Shadow Dexterous hand. We created five goal configurations for each task to evaluate different approaches. We report the mean and standard deviation of the normalized improvement (Q1). We show the quantitative results in Table 1, and the qualitative results in Figure 3.

We find that DexDeform is capable of completing the challenging long-horizon dexterous manipulation tasks, and significantly outperforms the compared baselines. On the challenging in-hand manipulation task **Flip**, we find that all baseline approaches fail to complete the task, while DexDeform is able to swiftly flip the wrapper into the air with fingertips and deform the dough. We hypothesize that the success of DexDeform comes from the ability to leverage the skill model for decomposing the high dimensional state space, which allows for efficient planning. On single-hand task **Folding**, we find that the BC agent would fold the dough in the wrong direction, while such behavior is not found for DexDeform. We hypothesize that this is because DexDeform is able to leverage the translational equivariance offered by the implicit representation during planning. DAPG agent is able to squeeze the dough and move it towards a location that best matches the general shape, but is unable to dexterously fold the dough over. PPO agent and TrajOpt agents are unable to squeeze or create meaningful shapes, and would slightly move the initial dough towards the target shape. On dual-hand task **Rope**, we find that DexDeform is able to match the shape with fine-grained details. The BC agent is able to generally match the shape, while DAPG, PPO, and TrajOpt agents fail to create meaningful shapes due to the high dimensional space created by two Shadow hands and two deformable objects. Due to the sample complexity of RL approaches, the speed of soft-body simulation limits the speed of convergence. We believe that with a large amount of samples, the performance of RL agents should improve, constituting a novel future direction.

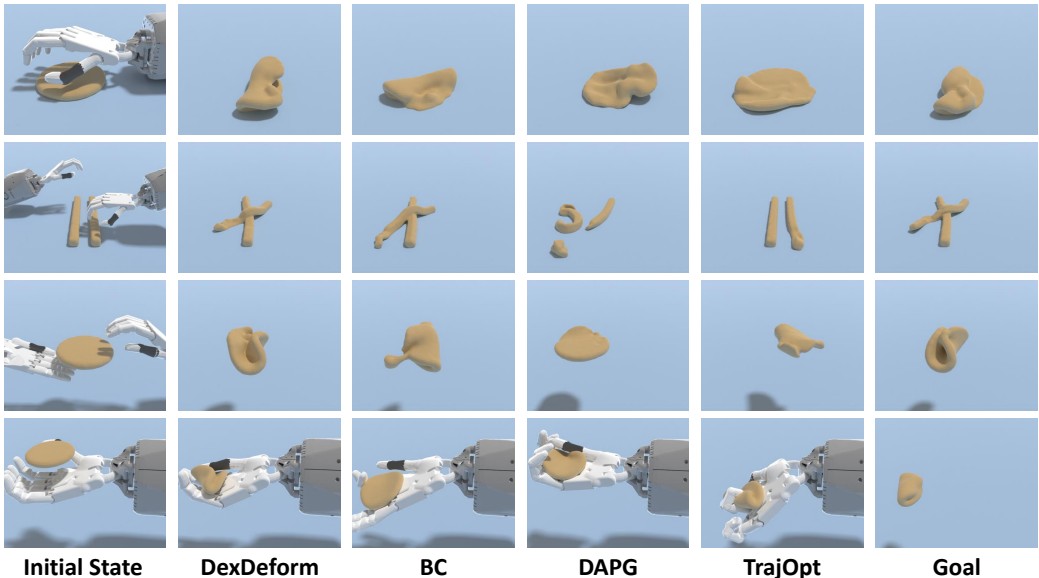

| Initial State | DexDeform | BC | DAPG | TrajOpt | Goal |

Figure 3: Qualitative results of each method on four environments: **Folding**, **Rope**, **Dumpling**, **Flip** (from top to bottom). The robot hand is not rendered for the first three environments to avoid occlusion of the final shape.

| **Env** | Folding | Rope | Bun |
|---|---|---|---|
| TrajOpt | $0.032 \pm 0.061$ | $0.079 \pm 0.026$ | $0.000 \pm 0.000$ |
| PPO | $0.361 \pm 0.173$ | $0.460 \pm 0.257$ | $0.069 \pm 0.117$ |
| DAPG | $0.538 \pm 0.308$ | $0.246 \pm 0.626$ | $0.460 \pm 0.079$ |
| BC | $0.685 \pm 0.388$ | $0.557 \pm 0.377$ | $0.379 \pm 0.258$ |
| DexDeform | $\mathbf{0.970 \pm 0.021}$ | $\mathbf{0.972 \pm 0.010}$ | $\mathbf{0.874 \pm 0.078}$ |
| **Env** | Dumpling | Wrap | Flip |
| TrajOpt | $0.000 \pm 0.000$ | $0.000 \pm 0.000$ | $0.195 \pm 0.275$ |
| PPO | $0.000 \pm 0.000$ | $0.000 \pm 0.000$ | $0.223 \pm 0.328$ |
| DAPG | $0.000 \pm 0.000$ | $0.000 \pm 0.000$ | $0.000 \pm 0.000$ |
| BC | $0.506 \pm 0.314$ | $0.134 \pm 0.595$ | $0.253 \pm 0.359$ |
| DexDeform | $\mathbf{0.888 \pm 0.055}$ | $\mathbf{0.845 \pm 0.050}$ | $\mathbf{0.842 \pm 0.057}$ |

Table 1: The averaged normalized improvements and the standard deviations of each method.

### 3.3 ABLATION ANALYSIS OF DEXDEFORM

To quantify the improvement brought by differentiable physics (Q2), we perform ablatively compare DexDeform against a baseline, named Skill-Only, that does not use differentiable physics for exploration. In a fashion similar to our previous table, we report the normalized improvement in Table 2. We find that the Skill-Only agent, trained entirely on the initial human demonstrations, is unable to generalize across evaluation goals that are uncovered by the initial dataset. Gradient-based trajectory optimization leveraged the interactions with the environment and exploited the gradient information to achieve fine-grained control over the soft-body shapes.

To evaluate the benefits of the skill model (Q3), we perform an ablation that compares DexDeform with a baseline (NN-TrajOpt) that replaces the skill model with a heuristic. Given a goal shape, NN-TrajOpt uses EMD to find the nearest neighbor of that shape from the initial human demonstration data. NN-TrajOpt then uses a gradient-based optimizer to refine the corresponding trajectory of this nearest neighbor. We report the qualitative comparison in Figure 4. We illustrate that pure EMD might not be a good measure for soft bodies with large topological variations Feydy (2020). In contrast, DexDeform leverages skill embedding and is able to compositionally represent the deformation process, allowing for finding the suitable policy.

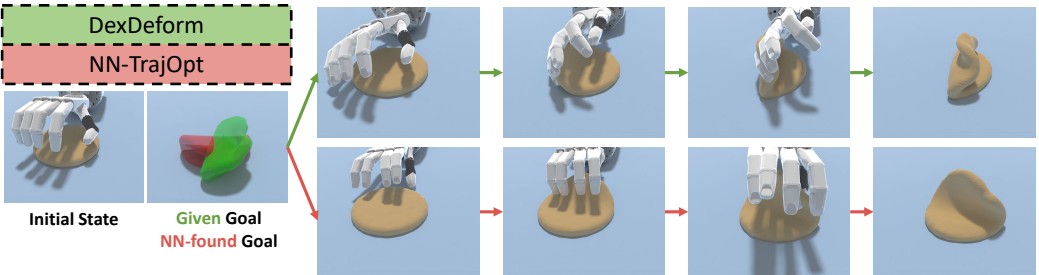

Figure 4: Ablative comparison with NN-TrajOpt that replaces skill model with a heuristic.

| Env | Folding | Rope | Bun | Dumpling |
|---|---|---|---|---|
| Skill-Only | $0.908 \pm 0.058$ | $0.914 \pm 0.023$ | $0.820 \pm 0.008$ | $0.725 \pm 0.244$ |
| DexDeform | $\mathbf{0.970 \pm 0.021}$ | $\mathbf{0.972 \pm 0.010}$ | $\mathbf{0.874 \pm 0.078}$ | $\mathbf{0.888 \pm 0.055}$ |

Table 2: Ablative comparison with Skill-Only that does not use the gradient-based optimizer. We show the averaged normalized improvements and the standard deviation of each method.

## 3.4 Skill Model Visualization

To see whether skill dynamics predictions are consistent with action rollouts (Q4), we visualize the latent encoding of the scene $\hat{s}_t$, predicted by the skill dynamics model given a skill embedding, as well as the ground truth state $S_t$ obtained from actions predicted by the skill decoder. Ideally, the two visualizations should show consistency. As shown in Figure 7, we observe a high level of consistency between the skill dynamics and the skill decoder. To find out what the latent space of skill embedding looks like (Q5), we visualize the skill embeddings using t-distributed stochastic neighbor embedding (t-SNE) (van der Maaten & Hinton, 2008) on **Folding**. We label each embedding based on the location of the final shape achieved by the corresponding skill sequence. We partition the ground plane into five parts: left, right, front, back, and center. As shown in Figure 8, the skill embeddings are correlated with the label categories. Details and visualizations can be found in Appendix E.

## 4 Related Work

**Dexterous Manipulation.** Dexterous manipulation has been a long-standing challenge in robotics, with the early works dating back to Salisbury & Craig (1982); Mason et al. (1989). Different from parallel-jaw grasping, dexterous manipulation typically continuously controls force to the object through the fingertips of a robotic hand (Dafle et al., 2014). There have been many prior works on using trajectory optimization (Mordatch et al., 2012; Bai & Liu, 2014; Sundaralingam & Hermans, 2019) or kinodynamic planning (Rus, 1999) to solve for the controllers. However, to make the optimization or planning tractable, prior works usually make assumptions on known dynamics properties and simple geometries. Another line of works uses reinforcement learning to train the controller. Some model-based RL works learned a dynamics model from the rollout data (Kumar et al., 2016; Nagabandi et al., 2020), and used online optimal control to rotate a pen or Baoding balls on a Shadow hand. OpenAI et al. (2020; 2019) uses model-free RL to learn a controller to reorient a cube and transfer the controller to the real world. To speed up the policy learning when using model-free RL, Chen et al. (2022) uses a teacher-student framework to learn a controller that can reorient thousands of geometrically different objects with both the hand facing upward and downward. (Radosavovic et al., 2020; Zhu et al., 2019; Rajeswaran et al., 2017; Jeong et al., 2020; Gupta et al., 2016; Qin et al., 2021) bootstraps the RL policy learning from demonstration data for reorienting a pen, opening a door, assembling LEGO blocks, etc. Handa et al. (2020); Arunachalam et al. (2022); Sivakumar et al. (2022) developed a teleoperation system for dexterous manipulation by tracking hand pose and re-targeting it to a robot hand. Unlike previous works with rigid bodies, our work performs the first investigation on the learning-based dexterous manipulation of soft bodies that carries infinite degrees of freedom, and provides a differentiable simulation platform for teleoperation.

**Learning Skills from Demonstrations.** Our skill model shares the same spirits with hierarchical imitation learning (Fang et al., 2019; Shi et al., 2022b; Gupta et al., 2019; Lynch et al., 2020) and motion synthesis (Peng et al., 2018; 2022), which view skill learning as sequential modeling tasks (Janner et al., 2021; Chen et al., 2021) in a low-dimensional space. Following Shi et al. (2022b), we learn a latent dynamic model to compose skills with model-based planning (Hafner et al., 2019; 2020) in the latent space. We employ these ideas for deformable object manipulation, where we integrate skill abstraction and latent dynamics into our pipeline. Our additional innovation is an exploration phase guided by the gradient-based trajectory optimizer, learning dexterous soft-body manipulation skills with a small number of demonstrations.

**Deformable Object Manipulation.** Deformable object manipulation have attracted great attention because of its wide range of applications in the real world. Previous works have explored manipulating different materials from objects humans interact with on a daily basis, including cloth (Maitin-Shepard et al., 2010; Hoque et al., 2021; Lin et al., 2021; Huang et al., 2022; Weng et al., 2021; Liang et al., 2019; Wu et al., 2020), rope (Sundaresan et al., 2020; Mitrano et al., 2021; Yan et al., 2020; Wu et al., 2020), and fluid materials (Ma et al., 2018; Holl et al., 2020; Li et al., 2022b; Schenck & Fox, 2017; Gautham et al., 2022). Our work is built upon Huang et al. (2020), which uses the MPM (Jiang et al., 2016) to simulate elastoplastic objects (Huang et al., 2020; Li et al., 2019; Shi et al., 2022a; Figueroa et al., 2016; Matl & Bajcsy, 2021; Heiden et al., 2021), and is able to represent materials such as dough and clay. Different from previous works, we investigate how to interact with deformable objects using multi-fingered hands, which carry versatility across different scenarios.

**Differentiable physics.** The development of differentiable simulator (Bern et al., 2019; Geilinger et al., 2020; Liang et al., 2019; Hu et al., 2019b;a; Huang et al., 2020; Qiao et al., 2021; Du et al., 2021; Heiden et al., 2019; Geilinger et al., 2020; Werling et al., 2021; Howell et al., 2022) enables fast planning (Huang et al., 2020), demonstration generation (Lin et al., 2022) and adaptation (Murthy et al., 2020). Systems have been developed to generate high-performance simulation code for the support of automatic differentiation (Hu et al., 2019a; Macklin, 2022; Freeman et al., 2021). However, many works have discovered that trajectory optimizers with first-order gradients are sensitive to local optima (Li et al., 2022a; Suh et al., 2022; Xu et al., 2022; Antonova et al., 2022). Many have found that the gradient-based optimizer can benefit from the integration of sampling-based methods, which enables global search to escape from local optima. The skill model employed by our method can be viewed as a form of planning. Different from previous methods, the skill model can decompose the high dimensional policy space, which enables efficient planning in the latent skill space.

## 5 CONCLUSION

In this work, we perform, to the best of our knowledge, the first investigation of the learning-based dexterous manipulation of deformable objects. We build a platform that integrates low-cost teleoperation with a soft-body simulation that is differentiable. We propose DexDeform, a principled framework that abstracts dexterous manipulation skills from human demonstrations, and refines the learned skills with differentiable physics. We find that DexDeform outperforms the baselines and accomplishes all six challenging tasks.

There are a few interesting directions for future work. With our simulation platform, it would be interesting to leverage the vast amount of in-the-wild videos (e.g., making bread, stuffing dumpling, building sand castle) for learning dexterous deformable manipulation policies in the future. It is also intriguing to speed up the soft-body simulation for large-scale learning with RL. Our work assumes full point cloud observation. Although our choice of implicit representation has been shown to transfer from the simulation into real-world robotic deployment by Shen et al. (2022), we would like to work with real-world observations in the future.

**Acknowledgement.** This project was supported by the DARPA MCS program, MIT-IBM Watson AI Lab, and gift funding from MERL, Cisco, and Amazon.

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

## A  Environment details

Here, we provide descriptions of each task covered in Figure 1. These descriptions (e.g., which hand to use first, where to grasp) are made based on how human demonstrations are collected. There could exist many other solutions for each task.

- **Folding** The agent is spawned above the dough wrapper, and needs to fold the wrapper to four general directions, including front, back, left, and right. The task length is 250 steps.
- **Wrap** The agent needs to first grasp and move the plasticine ball onto the rope, and then pinch on the side of the rope to wrap the ball over. The task length is 500 steps.
- **Flip** The agent needs to swiftly flip the dough wrapper into the air, and deform and reorient it with agility. The task length is 500 steps
- **Bun** The agent needs to use two hands to simultaneously pinch and push the wrapper into a bun-shaped object. The task length is 250 steps
- **Rope** The agent needs to use the right hand to grasp onto the rope on the right, lift and place it above the rope on the left. Finally, the left hand needs to bend the rope that was originally on the left. The task length is 250 steps.
- **Dumpling** The agent needs to first use the right hand to grasp onto the right side of the wrapper. While holding the dumpling with the right hand, the left hand needs to lift the left side. Finally, the two hands need to bring together the dough into a dumpling-shape object. The task length is 250 steps.

We build our simulation environments on top of PlasticineLab (Huang et al., 2021), a differentiable physics simulator based on the MLS-MPM algorithm (Hu et al., 2018).

For single hand environments, **Folding** and **Wrap** have an action dimension of 26 (20 for actuators for finger joints and wrists, and 6 for base). For in-hand manipulation environments, we assume a single hand with fixed base that has 20 action dimension. All dual hand environments have an action dimension of 52 with movable bases.

### A.1  Details on robot actions

The robot actions in our simulated environment correspond to the relative change in the joint angles on each actuated joint. I.e., $q_{t+1}^{joint} = q_t^{joint} + a_t$

### A.2  Details on point cloud observations

For the point clouds, we assume full point cloud observation of the scene from the MPM simulation, which includes the multi-fingered hand(s) and the object(s). We have also conducted experiments taking partial point clouds observed through four RGBD camera viewpoints as inputs. The experiment results and interpretations are reported in Appendix H.

## B  Details on teleoperation

Our teleoperation system is based on the Leap Motion Controller (Spiegelmock, 2013), an optical hand tracking device. By constructing an inverse kinematics model based on the Shadow hand, our system re-targets the detected human finger positions into the joint positions of the simulated robot hand that is controlled via a position-based PD controller. Our system runs teleoperation, simulation, and rendering with multiprocessing, and achieves 15-20 FPS on a laptop with NVIDIA GeForce RTX3070 Laptop GPU. Our system setup is illustrated below in Figure 5.

## C  Details on skill model training and planning

### C.1  Implicit Scene Representation

**ConvONet Encoder.** A shallow PointNet encoder Qi et al. (2017) first maps the three-dimensional input coordinates and their rgb colors into a feature space. The PointNet encoded features are then

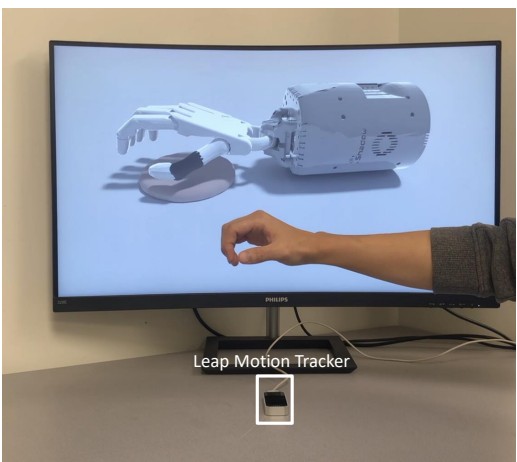

Figure 5: The hardware setup of our teleoperation system with the Leap Motion Tracker with a human operator. The computer is hidden for better illustration of the system.

projected orthographically onto three canonical planes that represent xz, xy, and yz planes. The triplane 2D feature grids are then passed into a U-Net Çiçek et al. (2016). Concretely, we choose to use triplane resolution of 128 with hidden dimension equals to 64, and the depth of U-Net is 5.

**ConvONet Decoder.** Since the triplane feature grids represent a parameterization of the scene, we can obtain the occupancy prediction (empty space, hand, or object) at any query point. The implicit scene decoder first uses bilinear interpolation to retrieve the local feature of the query point from each of the triplanes. The interpolated features from different planes are then summed and passed to a multi-layer perceptron (MLP) to decode into the final occupancy prediction. In practice, we use a 5-layer MLP with a hidden dimension of 32.

**Latent Encoding of the Scene.** We use a convolutional VAE to encode the triplanes from ConvONet encoder into a latent encoding, and to decode a latent code back into the space of implicit scene representations. Concretely, the VAE has an encoder with 6 convolutional layers with a kernel size of 3 and stride of 2, and channel size of (64, 64, 128, 256, 512, 512), followed by an MLP that maps the feature into a latent code of 128 dimensions.

**Training Details.** We train the ConvONet model for $10,000$ iterations, and pretrain the latent scene VAE for $50,000$ iterations.

## C.2 SKILL MODEL

**Input State-Action Sequence.** We use $K = 10$ in practice, which is the abstraction length. During training, each sample is a sequence of 50 consecutive state-action pair, which can be encoded into 5 skill abstractions.

**Skill Encoder.** We use a 5 layer LSTM Hochreiter & Schmidhuber (1997) with dimension of 128 to encode the input sequence of state-action pairs. Following the VAE training paradigm, the encoded feature then passed through a MLP that maps the feature into a latent code of 128 dimensions.

**Skill Dynamics Predictor.** The dynamics predictor is a 3-layer MLP with dimensions (512, 256, 128).

**Skill Action Decoder.** The action decoder is a 3-layer MLP with dimensions (512, 256, act_dim), Where the action dimension depends on whether the task uses a single hand or dual hands.

**Training Details.** Following SkiMo (Shi et al., 2022b), we jointly train the skill encoder, dynamics predictor, and action decoder with the following objectives for every skill embedding $\mathbf{z}$. We first present the objective function for the skill encoder and skill action decoder trained under the VAE setting.

$$\mathcal{L}_{\text{sVAE}} = \mathbb{E}_{(s,a)_{1:K} \sim \mathcal{D}} \left[ \frac{1}{K} \sum_{i=1}^{K} (\pi(z, s_i) - a_i)^2 + \beta \cdot KL(f(z|(s,a)_{1:K} \| p(z)) \right]. \quad (2)$$

Let $N = 5$ be the number of skill abstractions obtained from each sample that is of 50 steps. We show the objective function for skill dynamics

$$\mathcal{L}_{\text{Dyn}} = \mathbb{E}_{(s,a)_{1:K} \sim \mathcal{D}} \left[ \sum_{i=0}^{N-1} \left\| \mathcal{T}(z_{iK+1}, s_{iK+1}) - s_{(i+1)K+1} \right\|_2^2 \right]. \quad (3)$$

Combining the objectives above, we jointly train the skill encoder, dynamics predictor, and action decoder.

$$\mathcal{L}_{\text{skill}} = \mathcal{L}_{\text{sVAE}} + \mathcal{L}_{\text{Dyn}} \quad (4)$$

**Planning Details.** Given a target shape, we iteratively plans actions using the optimization problem described in Equation 1. We apply the following algorithm to plan and execute actions. The initial batch of skill sequences is sampled from the standard Gaussian distribution.

---

**Algorithm 1** Planning actions with skill model

---

**Input:** Goal shape $g$, approximate task horizon $H$, abstraction length $K$, number of abstractions to execute each time $M$
1: remain $\leftarrow \lfloor H/K \rfloor$
2: **while** remain $> 0$ **do**:
3:     Sample batch of $\{z_1, z_K, ..., z_H\}_{j=1}^J$ from standard Gaussian to initialize;
4:     Optimize batch of $\{z_1, z_K, ..., z_H\}_{j=1}^J$ according to Eqn. 1 and choose the best sequence;
5:     Use $\pi$ to decode the first $M$ skills into $MK$ actions to execute;
6:     remain $\leftarrow$ remain $- M$;
7: **end while**

---

## D    DETAILS OF DEXDEFORM

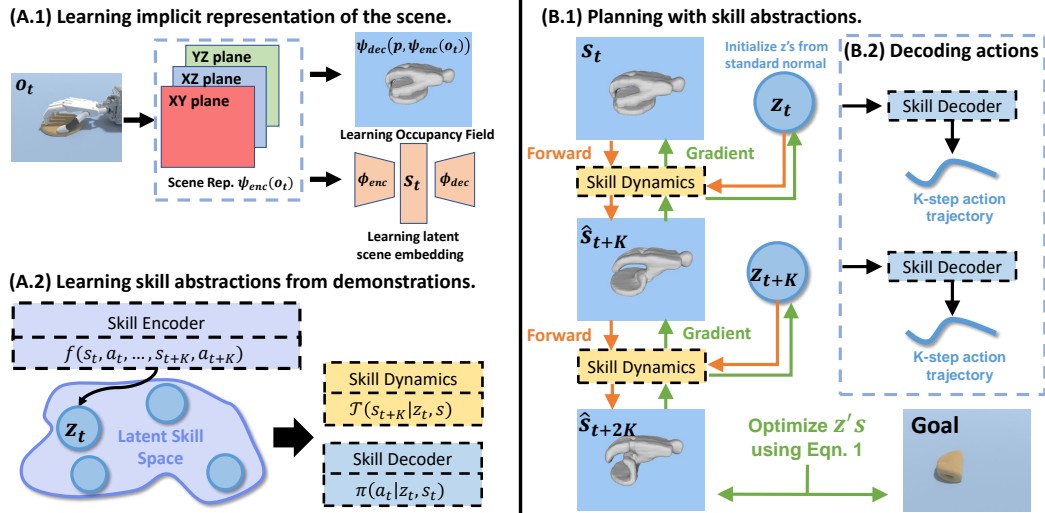

Figure 6: An overview of **training (A)** and **inference (B)** in DexDeform. Demonstration collection using teleoperation and differentiable physics guided exploration are omitted.

The DexDeform algorithm presented in Algo. 2, contains the training phase and exploration phase. The training phase involves learning implicit representation of the scene and the skill model, while the exploration phase leverages differentiable physics optimizer for expanding the demonstration set

to new goals. We start with a collection of human demonstrations and iteratively execute the training phase and the exploration phase to explore novel goals. In practice, for each iteration, we generate one new trajectory from each original human demonstration based on goal augmentations. In Fig. 6, we illustrate the details of training and inference procedures in DexDeform.

---

**Algorithm 2** Iterative learning with DexDeform

---

**Input:** human demonstrations, number of iterations $N$, new trajectories per iteration $M$.
 1: Initialize dataset $\mathcal{D}$ with human demonstrations;
 2: **for** $i \leftarrow 1$ **to** $N$ **do**
 3:     Train implicit representations and skill model $f, \mathcal{T}, \pi$ using $\mathcal{D}$ until convergence;
 4:     $m \leftarrow 0$;
 5:     **while** $m < M$ **do**:                                                 $\triangleright$ Exploration Phase
 6:         Sample a goal from $\mathcal{D}$ and apply augmentation to get $\tilde{\mathbf{g}}$;
 7:         Optimize batch of $\{z_1, z_K, ..., z_H\}_{j=1}^{J}$ according to Eqn. 1 and choose the best sequence;
 8:         Use $\pi$ and Algo. 1 to decode skill sequence into trajectory $\tau = \{o_1, a_1, ..., o_H, a_H\}$;
 9:         Refine the trajectory with differentiable physics to get $\tilde{\tau}$;
10:         **if** $D_{emd}(\tilde{o}_H, \mathbf{g}) < \epsilon$ **then**           $\triangleright$ If the shape difference threshold is satisfied
11:             Store $\tilde{\tau}$ in dataset $\mathcal{D}$;
12:             $m \leftarrow m + 1$;
13:         **end if**
14:     **end while**
15: **end for**

---

## E   Skill Visualization Details

### E.1   Skill Dynamics

In Figure 7, to create the visualizations of $\hat{s}_t$, we decode the latent encoding of the scene into 2D feature grids. Since the feature grids represent an implicit parameterization of the scene with occupancy information, we apply Multiresolution IsoSurface Extraction (MISE) (Mescheder et al., 2019) to extract meshes of the scene predicted in imagination.

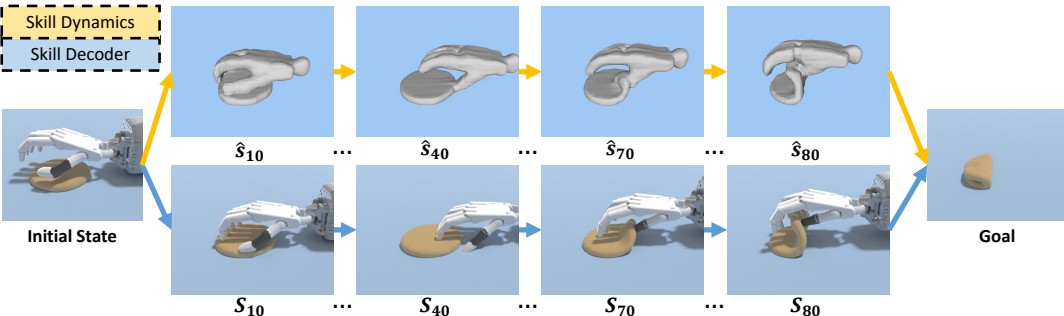

Figure 7: We visualize the skill dynamic predictions on **Folding**, and show consistency between the predicted state and the resulting states from actions predicted by the skill decoder.

### E.2   Tsne Visualization

We visualize the skill embeddings using t-distributed stochastic neighbor embedding (t-SNE) (van der Maaten & Hinton, 2008) on **Folding**. We label each embedding based on the location of final shape achieved. We partition the ground plane into five parts: left, right, front, back, and center. As shown in Figure 8, we see that the skill embeddings are correlated with the label categories. Our t-SNE is ran with 1 million iterations.

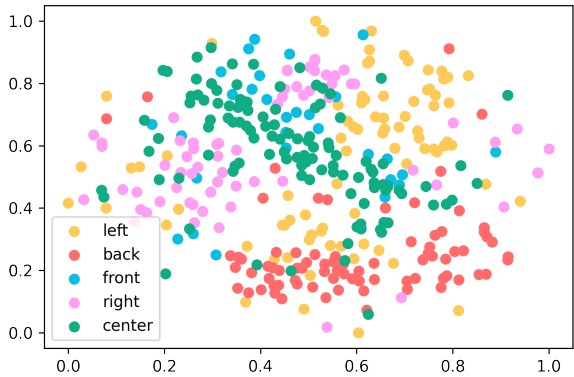

Figure 8: Skill Tsne Visualization on **Folding**

## F    RESULTS ON DEXDEFORM WITH PARTIAL POINT CLOUD

We have conducted experiments taking partial point clouds observed through four RGBD camera viewpoints as inputs. The four cameras are placed on the front, back, left, and right sides of the tabletop (ground plane). We report the averaged normalized improvements in Table 5.

| Env | Folding | Rope | Bun | Dumpling | Wrap | Flip |
|---|---|---|---|---|---|---|
| DexDeform w/ perfect pointcloud | **0.970** | **0.972** | **0.874** | **0.888** | **0.845** | **0.842** |
| DexDeform w/ partial pointcloud | 0.954 | 0.961 | 0.814 | 0.785 | 0.809 | 0.683 |

Table 3: The averaged normalized improvements of each method.

In general, we find that our use of implicit scene representation allows for generalization across partial point clouds with a varying number of points observed in each frame.

## G    RESULTS ON BASELINES WITH LATENT SCENE EMBEDDING

We have conducted experiments where the baseline models use the same latent embeddings as our proposed methods. We report the averaged normalized improvements of each approach in Table 4. For convenience, we have included the original baselines and our approach.

| Env | Folding | Rope | Bun | Dumpling | Wrap | Flip |
|---|---|---|---|---|---|---|
| PPO w/o latent | 0.361 | 0.460 | 0.069 | 0.000 | 0.000 | 0.223 |
| PPO w/ latent | 0.630 | 0.531 | 0.000 | 0.100 | 0.082 | 0.270 |
| DAPG w/o latent | 0.538 | 0.246 | 0.460 | 0.000 | 0.000 | 0.000 |
| DAPG w/ latent | 0.459 | 0.544 | 0.294 | 0.272 | 0.183 | 0.000 |
| BC w/o latent | 0.685 | 0.557 | 0.379 | 0.506 | 0.134 | 0.253 |
| BC w/ latent | 0.672 | 0.524 | 0.000 | 0.145 | 0.433 | 0.326 |
| DexDeform | **0.970** | **0.972** | **0.874** | **0.888** | **0.845** | **0.842** |

Table 4: The averaged normalized improvements of each method.

In general, we find that the inclusion of latent scene embeddings did not significantly improve the performance of the baseline methods. We speculate that the policy characteristics of the baselines are more responsible for the performance than the designs of their perception modules.

Specifically, for the RL approaches (PPO & DAPG), the sample complexity becomes a limitation under dexterous deformable object manipulation due to the large state space. Behavior cloning

struggles to generalize across different goals. Because BC lacks the ability to compositionally represent and reason about the deformation process, which is an advantage of skill-based planning used by our method.

## H    RESULTS ON SKILL MODEL WITH A LEARNED PRIOR

We have conducted additional experiments where a prior network $f_p(z_t|h_t, a_t, ..., h_{t-K}, a_{t-K})$ is learned jointly with the posterior network $f_q(z_t|h_t, a_t, ..., h_{t+K}, a_{t+K})$ (i.e., the skill encoder) using KL divergence. Concretely, we implemented the prior network using an LSTM architecture. During our iterative planning (Algo. 1), we maintain a 10-step history of past state embeddings and actions from executing $M$ skill latents in the previous planning iteration. For the current planning iteration, we use the learned prior to sample the batch of future skill latents for initializing the optimization in Eqn. 1. For the initial planning iteration, we use the standard Gaussian prior to sample the skill latents for optimization.

We observed that the two approaches arrive at similar performances. We think it is an intriguing future direction to incorporate a temporal skill prior and consider other forms of exploration for tackling deformable object manipulation.

| Env | Folding | Rope | Bun | Dumpling | Wrap | Flip |
|---|---|---|---|---|---|---|
| DexDeform w/o learned prior | **0.970** | **0.972** | 0.874 | **0.888** | 0.845 | **0.842** |
| DexDeform w/ learned prior | 0.933 | 0.739 | **0.907** | 0.814 | **0.881** | 0.773 |

Table 5: The averaged normalized improvements of each method.

## I    DETAILS ON BASELINE EXPERIMENTS

We describe the input type, training (if applicable), and runtime usage of each baseline method below.

**Trajectory Optimization (TrajOpt).**

- **Input type**: This method takes the full state of the simulation as the observation at each timestep.
- **Runtime usage**: TrajOpt is not learning-based. TrajOpt uses first-order gradients from a differentiable simulator to solve for an open-loop action sequence Kelley (1960).

**Model-free Reinforcement Learning (PPO)**

- **Input type**: Our PPO agent takes goal-conditioned input. Concretely, suppose that $W_t^{scene}$ is the point cloud observation of the scene at timestep $t$, and $W^{goal}$ is the point cloud of the goal shape. The input to these baselines at timestep $t$ is the goal-conditioned point cloud $W_t = W_t^{scene} \cup W^{goal}$. For each point in the concatenated point cloud, a binary label indicates whether the point belongs to the goal.
- **Training**: We use a batch size of 400 and a learning rate of $3 \times 10^{-4}$ to train the agent for 3M steps or till convergence.
- **Runtime usage**: Given a goal specified in a point cloud, we create the goal-conditioned input at each timestep to perform a rollout using the task-specific horizon.

**Behavior Cloning (BC)**

- **Input type**: Same as PPO described above, our BC agent takes goal-conditioned point clouds as inputs.
- **Training**: We use a batch size of 128 and a learning rate of $3 \times 10^{-4}$ to train the agent for 3M steps or till convergence.
- **Runtime usage**: Given a goal specified in a point cloud, we create the goal-conditioned input at each timestep to perform a rollout using the task-specific horizon.

**Model-free Reinforcement Learning with Data Augmentation**   We compare against Demo Augmented Policy Gradient (DAPG) from Rajeswaran et al. (2017), a method that combines behavior cloning and online RL.

- **Input Type**: Same as PPO described in above, our DAPG agent takes goal-conditioned point clouds as inputs.
- **Training**: We initialize the policy with behavior cloning described above in Part C. We then apply RL finetuning to the initialized policy using PPO. The DAPG objective in Rajeswaran et al. (2017) (Eq. 6) involves two hyperparameters $(\lambda_0, \lambda_1)$ for the weighting function. We use $\lambda_0 = 0.1, \lambda_1 = 0.995$. We finetune for 3M steps or till convergence, using a batch size of 400 and a learning rate of $3 \times 10^{-4}$.
- **Runtime**: Given a goal specified in a point cloud, we create the goal-conditioned input at each timestep to perform a rollout using the task-specific horizon.

