# OpenReview forum: "DexDeform: Dexterous Deformable Object Manipulation with Human Demonstrations and Differentiable Physics"
_ICLR.cc/2023/Conference — ICLR 2023 poster_

### Official Review · Reviewer_WWJM · 2022-10-23

**Confidence:** 4
**Correctness:** 3
**Technical Novelty And Significance:** 2
**Empirical Novelty And Significance:** 3
**Recommendation:** 6

**Clarity, Quality, Novelty And Reproducibility:**

The paper could do with some improvement of clarity, particularly around the method for this to be reproducible. I would recommend a bit of a rewrite for clarity, particularly focusing on notation, and explaining the method (perhaps rethinking what is in the appendix and what isn't). This also applies to the abstract, which doesn't really convey the approach very clearly.

The paper proposes an interesting set of deformable object manipulation tasks and a method of learning these, which could be useful for others to build on and test their algorithms. The method itself does not seem to be particularly novel.

**Strength And Weaknesses:**

Strengths:

The proposed set of benchmarks is interesting, and the results appear to show that the proposed approach performs well at these.

Weaknesses:

The biggest concern I have here is that the description of the proposed approach is rather unclear, and I struggled to follow section 2.4. It is not at all clear to be how the different components in the model connect, and how the baseline approach is trained or used at run-time. It would be helpful to provide a figure illustrating this, and to standardise notation to align with Fig 2. Notation needs a lot of care, for example Locc is seemingly also referred to L(g,h) on page 4, and this makes it hard to keep up with and follow much of the method. I appreciate that some of this is in the appendix, but feel section 2.4 could do with a bit of a rewrite in service of the reader, as at present the complexity here limits reproducibility and does not do the method justice.


Minor points:

Par 2: "...consider the problem of deformable object manipulation with the Shadow Dexterous hand." please rephrase to a *simulated* shadow dexterous hand.

The paper repeatedly claims to provide the first dexterous manipulation benchmark for deformable objects (to the best of the authors knowledge), but learning based approaches have been widely used in robotics literature, along with many deformable manipulation tasks (rope, cables, cloth folding). Plasticinelab, which this work builds on, requires the manipulation of deformable objects. Is the definition of dexterous manipulation used here specifically down to the use of a multi-fingered hand?

Can you provide additional detail about the environments? Eg. actions (are these joint torques?) and point clouds (are these rendered with respect to the camera view point? do they factor in occlusions) etc.

Section 2.3 "Given a novel deformable shape, how do we solve it..." Rephrase - I don't think you can solve a shape.

Section 3.3: "We illustrate that pure EMD might not be a good measure for soft bodies with large topological variations Feydy (2020)" - I don't see an illustration of this. If EMD is not a good metric, shouldn't the benchmark report alternatives too?


**Summary Of The Paper:**

This paper explores the task of dexterous manipulation with human demonstrations. The paper introduces an interesting set of simulated benchmarking tasks, along with a baseline approach to solve these. The benchmark builds on prior work (Plasticinelab), using a simulated shadow robot hand. The proposed baseline relies on human demonstrations captured using a leap motion controller, and mapped to shadow hand using an ik solver. An implicit representation of the scene (observed in point cloud form) is learned using a point cloud encoder decoder model. this representation is then combined with actions, and a sequence of these mapped into another latent space, a dynamics model is learned using this representation, and used for goal-based planning.

**Summary Of The Review:**

This paper introduces an interesting benchmark for goal oriented dexterous deformable object manipulation tasks, alongside a latent learning-based planning approach. The explanation of the latter needs to be improved.

---- Post rebuttal updates ----

I think the authors have added sufficient clarity and addressed most of my concerns, hence am raising my score.

---

> ### Author Response · Authors · 2022-11-17
> **Response to Reviewer WWJM (1/3)**
>
> Thank you so much for your interest in our work and your constructive comments. We provide clarifications and answers to each of your concerns below.
>
>
>
> > #### 1. "The description of the proposed approach is rather unclear. How do different components in the model connect? How is the proposed approach trained or used at runtime? It would be helpful to provide a figure illustrating this, and to standardise notation to align with Fig 2."
>
> Thanks so much for your helpful writing suggestions to strengthen our work. We would like to address each part of your concern below.
>
>
> #### **[Connections among model components]**
>
> Our framework consists of three components. We describe these components' connections by marking each component's input and output.
>
> - (Step 1, Sec. 2.1) Using *human teleoperation* (**S1 input**), We collect *a small set of demonstrations* (**S1 output**) for each task.
> - (Step 2, Sec. 2.2) We use the *collected demonstration* (**S2 input**) to train the *skill model* (**S2 output**). Given a target deformable shape, we use the skill model to plan a sequence of actions to achieve the target shape.
> - (Step 3, Sec. 2.3) More interactions are needed for the skill model to generalize across novel goals that are not captured in the initial demonstration set. We use shape augmentation to sample novel goals. Next, we use the *trained skill model* (**S3 input**) to plan a sequence of actions for the sampled novel goals. We employ gradient-based optimization to refine the action sequences and store *the successful trajectories as new demonstrations* (**S3 output**).
> - Since **Step 2 and Step 3 form a loop**, we can repeat them to **iteratively finetune the skill model on the newly expanded goals**.
>
> The pipeline above is illustrated in Figure 2. We have brought Algorithm 2 (now Algo. 1) for the procedure above over to Sec. 2.3, and reorganized the structure of Sec. 2.3 to improve clarity.
>
>
> #### **[Training and runtime inference of the proposed method (DexDeform)]**
>
> We have followed the reviewer's suggestion and provided an additional Figure 6 in Appendix D that illustrates the training and inference processes of our proposed method.
>
> ##### A. Training
>
> The training of DexDeform follows our description above and is described in Algorithm 1 in our revised paper. Specifically, this procedure takes **initial human demonstration**, **the number of training iterations $N$**, and **the number of new trajectories to create per iteration $M$** as inputs. For each iteration, we execute **Step 2** to train the implicit representation of the scene and the skill model. We then execute **Step 3** to generate $M$ new trajectories with novel goals. **The end result of training is a skill model that generalizes across different goal shapes.**
>
> ##### B. Runtime Inference
>
> Given a target deformable shape, we use the skill model to plan a sequence of actions to achieve the target shape. The planning procedure is described in Algorithm 2 in our revised paper. The planning objective is described in Equation 1.
>
> Specifically, we optimize a batch of skill latent sequences $\{z_1, z_K, ..., z_H\}^{J}_{j=1}$ according to the optimization objective Eqn. 1. **Using the skill dynamics predictor**, this objective aims to find a sequence of skills such that **the final predicted implicit scene embedding**, when decoded, **is occupied by the points in the target shape under the object category**. We choose the best skill sequence from a batch that achieves the lowest cost in the planning objective. Finally, **using the skill action decoder**, we decode the planned skill latents into actions executed on the robot hand.
>
> #### **[Notation and Typos]**
> Thank you for the detailed comments! We have standardized our notation throughout the paper, and aligned our notation with Fig. 2. We have also fixed the typo on $L_{occ}$ you pointed out.
>
>
>
>
> ---
>
> > #### 2. Rephrase the problem statement to "a *simulated* shadow dexterous hand". Rephrase *"Given a novel deformable shape, how do we solve it..."*.
>
> Thank you for the suggestion. We have rephrased these two sentences.

---

> > ### Author Response · Authors · 2022-11-17
> > **Response to Reviewer WWJM (2/3)**
> >
> > > #### 3. "The paper claims to provide the first dexterous manipulation benchmark for deformable objects, but learning based approaches have been used in deformable manipulation tasks. Is the definition of dexterous manipulation used here specifically down to the use of a multi-fingered hand?"
> >
> > #### **[Definition of dexterous manipulation in our work]**
> > Dexterous manipulation typically refers to the continuous control of force on the object through a multi-fingered robotic hand. This convention is supported by early works such as Salisbury & Craig (1982) [1] and Mason et al. (1989) [2]. As stated in our abstract and introduction, **in this work, we aim to learn dexterous manipulation of deformable objects using multi-fingered hands**.
> >
> > #### **[Prior works in deformable object manipulation]**
> >
> > There indeed exist learning-based approaches for deformable object manipulation, such as Sundaresan et al. (2020) [3] for rope manipulation and Lin et al. (2021) [4] for cloth manipulation. But we highlight that **these methods couldn't easily transfer to dexterous manipulation**, and **contains design choices that are specific to their problem settings**. For instance, Lin et al. (2021) nicely tackle the problem of cloth smoothing, where actions are parameterized as pick-and-place points for simplification, and the reward function computes the area covered by the cloth on the ground plane. However, these two design choices do not fit our problem setting, since we use a multi-fingered hand and care about shape matching in elastoplastic objects.
> >
> >
> > [1] Salisbury, J. Kenneth, and John J. Craig. "Articulated hands: Force control and kinematic issues." The International journal of Robotics research 1.1 (1982): 4-17.
> >
> > [2] Mason, Matthew T., and J. Kenneth Salisbury Jr. "Robot hands and the mechanics of manipulation." (1985).
> >
> > [3] Sundaresan, Priya, et al. "Learning rope manipulation policies using dense object descriptors trained on synthetic depth data." 2020 IEEE International Conference on Robotics and Automation (ICRA). IEEE, 2020.
> >
> > [4] Lin, Xingyu, et al. "Learning visible connectivity dynamics for cloth smoothing." Conference on Robot Learning. PMLR, 2022.
> >
> > ---
> >
> > > #### 4. Additional details about the environments (e.g., actions and point clouds).
> >
> > Thank you for suggesting providing additional details about our environments. We describe them below and have added these details in Appendix A.
> >
> > #### **[Details on robot actions]**
> >
> > The robot actions in our simulated environment correspond to the relative change in the joint angles on each actuated joint. I.e., $q_{t+1} = q_{t} + a_{t}$.
> >
> > #### **[Details on point cloud observations]**
> > For the point clouds, we assume full point cloud observation of the scene from the MPM simulation, which includes the multi-fingered hand(s) and the object(s). To further strengthen our work, we have conducted experiments taking partial point clouds observed through four RGBD camera viewpoints as inputs. The experiment results are included below and in Appendix F, where we report the averaged normalized improvements.
> >
> > | Env      | Folding       |   Rope   | Bun     |  Dumpling | Wrap | Flip |
> > | ---------| ------------- | -------- | ------  | --------- | ---- | ---- |
> > | Ours w/ Perfect point cloud |  **0.970**   |  **0.972** | **0.874** | **0.888**  | **0.845** | **0.842** |
> > | Ours w/ Partial point cloud |  0.954   |  0.961 | 0.814 | 0.785  | 0.809 | 0.683 |
> >
> >
> > In general, we find that our use of implicit scene representation **allows for generalization across partial point clouds** with **a varying number of points observed in each frame**.
> >
> >
> > While we think that it is a **crucial future direction to build perception modules that tackle other challenges** (e.g., using just a single-view camera, inferring physical properties, tackling hand occlusions), we would like to point out that our work offers a **nearly full-stack solution** to the challenging problem - learning dexterous manipulation of deformable objects. Our work provides **a simulation platform** that is fully differentiable with **a teleoperation system**, and **a novel framework** to learn perception and actions that generalize across partial point clouds.

---

> > > ### Author Response · Authors · 2022-11-17
> > > **Response to Reviewer WWJM (3/3)**
> > >
> > > > #### 5. Question about authors' statement: "we illustrate that pure EMD might not be a good measure for soft bodies with large topological variations". "I don't see an illustration of this. If EMD is not a good metric, shouldn't the benchmark report alternatives too?"
> > >
> > > We clarify that there exist two metric spaces that could be useful to evaluate EMD. The first one measures **the distance it takes to physically deform shape $A$ into shape $B$**, whereas the second one measures the **pure geometric differences between the two shapes**. The first metric space is relatively informal and used for providing intuition.
> > >
> > > When we state that "pure EMD might not be a good measure for soft bodies with large topological variations," **we mean that the physical process of deformation is not easily considered by pure EMD**, **in reference to the first metric space**.
> > >
> > > To further clarify the context of our statement, in our ablation analysis (Section 3.3), we are comparing against a trajectory optimization approach (*NN-TrajOpt*). *NN-TrajOpt* selects a trajectory from the existing demonstration data to initialize the optimization process. Given a target shape, *NN-TrajOpt* selects an initial trajectory that contains a shape that EMD considers to be the nearest neighbor of the target shape.
> > >
> > > As expected, since EMD cannot accurately account for the physical process of deforming a shape during planning (i.e., choosing the initial trajectory), ***NN-TrajOpt* chooses the optimization starting point based on strong geometric analysis**. By contrast, **DexDeform takes the physical interactions into account during planning by using a skill-based dynamics predictor**. This comparison is illustrated in Figure 4.
> > >
> > >
> > > ---
> > >
> > > **We hope that our response has addressed your concerns and turned your assessment to the positive side.**  *Please do not hesitate to contact us if there are other clarifications or experiments we can offer.*
> > >
> > > Best, Authors

---

> ### Author Response · Authors · 2022-11-23
> **Looking forward to your reply**
>
> Dear Reviewer WWJM,
>
> Happy Thanksgiving!
>
> Thank you again for your time. As the deadline for discussion is approaching, we hope to have a further conversation with you to see if our response resolves your concerns. We are happy to provide any additional clarifications that you may need.
>
> We would appreciate you replying to the most critical points in our response regarding writing clarity. We explained the connections among different model components, described the training and inference processes of our proposed method, and standardized our notation throughout the paper.
>
> We would appreciate you kindly checking our response. Please do not hesitate to contact us if there are further clarifications or experiments we can offer. Thanks!
>
> Best wishes, Authors

---

> > ### Comment · Reviewer_WWJM · 2022-11-23
> > **Apologies for the delayed response**
> >
> > Thank you for the detailed response to my review - I really appreciate this, and am willing to increase my score.

---

> > > ### Author Response · Authors · 2022-11-23
> > > **Thank you**
> > >
> > > Thank you so much for your helpful and constructive comments. We hope to improve the score further. Please don't hesitate to contact us if there are clarifications or experiments we can offer. Thank you again for your time.
> > >
> > > Best, Authors

---

### Official Review · Reviewer_whUB · 2022-10-24

**Confidence:** 3
**Correctness:** 3
**Technical Novelty And Significance:** 3
**Empirical Novelty And Significance:** 3
**Recommendation:** 8

**Clarity, Quality, Novelty And Reproducibility:**

The paper is well-written. It is clear to understand the problem, motivation, and contributions of the paper. The task and components of the pipeline are novel.

**Strength And Weaknesses:**

Strengths:

1- The paper is well-motivated. The concerns raised at the beginning are later addressed.

2- The optimization objective in Eq. 1 leverages the implicit scene representation, nicely overcoming the challenges of modeling soft objects.

3- Few demonstrations are enough to initialize the models, and it is straightforward to gather them by using the proposed teleoperation setup.

---

Weaknesses:

1- What is the design choice behind using a non-dynamic skill abstraction network? I think a temporal variant with a learnable prior would be more useful and allow for sampling of the next latent sample z. A sampled latent trajectory could be used in the exploration phase instead of augmenting the targets and running optimization in the observation space.

2- It is not clear if the prior of the skill abstraction network is used. I assume the latent skill samples in Eq. 1 are randomly initialized from the standard Gaussian prior. Could the authors clarify this?

3- I think the evaluations could be fairer. The baseline models take the input from the scene in point cloud representation. The implicit scene representation learned by the ConvONet and the scene embeddings learned by the VAE are trained in isolation by using standard training objectives. I believe the baseline models could use the same latent embeddings as the proposed method.

4- This is in line with my previous comment. The baselines and how they are implemented seem rather straightforward. A goal-conditioned policy (see [1*] for grasping rigid objects as an example) could be well suited for the tasks introduced in this paper. The goals could be represented by latent scene embeddings. This is not a weakness per se, but a suggestion. I think stronger baselines would improve the quality of the submission.

[1*] Christen, Sammy, et al. "D-Grasp: Physically Plausible Dynamic Grasp Synthesis for Hand-Object Interactions." Proceedings of the IEEE/CVF Conference on Computer Vision and Pattern Recognition. 2022.

**Summary Of The Paper:**

The paper tackles the task of dexterous manipulation of deformable objects and introduces a pipeline involving gathering human demonstrations, learning representations for the scene and the demonstrations, and augmenting novel data samples via a differentiable simulator. The proposed pipeline, DexDeform, starts with a small set of demonstrations to learn scene representations and skill abstractions, a dynamics model, and an initial policy. It then iteratively runs training and exploration stages where a gradient-based trajectory optimization is applied via a differentiable simulator to augment available samples. The paper presents evaluations on a variety of tasks, including soft object folding, wrapping, and flipping with a single or dual Shadow hand.

**Summary Of The Review:**

Despite the weaknesses I mentioned, I think it is an overall good paper. It provides useful insights into the problem as well as intriguing solutions. I am leaning toward accepting this paper. I'm looking forward to getting further clarification and experimental evidence from the authors.

-----
**Post rebuttal update:** The rebuttal addresses my concerns, and I do not have any further questions. I think this is an interesting paper for the community. I am in favor of accepting this paper. I raised my score from 6 to 8.

---

> ### Author Response · Authors · 2022-11-07
> **Looking for clarification**
>
> Thank you so much for your positive comments and constructive suggestions. Could you please clarify **Weakness 1**? If you question whether our skill abstraction network is designed to consider physical dynamics, the answer is yes.
>
> Specifically, the skill dynamics predictor takes a skill embedding and a scene embedding as inputs, and predicts the future scene embedding that is $K$ steps away in imagination. By optimizing a sequence of skill embeddings (initially sampled from the standard Gaussian prior) using the dynamics predictor, our planning objective in Eq. 1 takes physical dynamics into account.

---

> > ### Comment · Reviewer_whUB · 2022-11-09
> > **Thanks for the clarification**
> >
> > Thanks for the clarification.
> >
> > This is actually not a weakness. I was just curious if the authors had considered a probabilistic setting. More specifically, I meant a learned prior instead of the standard Gaussian one. The KL-Divergence term in Eq. 2 in section C.2 becomes $KL(f_q(z_t | h_t, a_t, ... , h_{t+K}, a_{t+K}) || f_p(z_t | h_t, a_t, ..., h_{t-K}, a_{t-K})$ where $f_q$ and $f_p$ correspond to the approximate posterior and learned prior, respectively. They are two separate networks with different sets of inputs. In this formulation, the prior learns to be the skill dynamics network $\tau$. It is part of the VAE architecture. For more details, you can take a look at the work of Chung, Junyoung, et al. "A recurrent latent variable model for sequential data." Advances in neural information processing systems 28 (2015).
> >
> > Considering the optimization of the randomly initialized latent samples in Eq. (1), the proposed skill encoder and the dynamics model also seem useful.

---

> > > ### Author Response · Authors · 2022-11-17
> > > **Thank you for the clarification**
> > >
> > > We thank the reviewer for the clarification. We conducted additional experiments where a prior network $f_{p}(z_t |h_t, a_t, ..., h_{t-K}, a_{t-K})$ is learned jointly with the posterior network $f_{q}$ (i.e., the skill encoder) using KL divergence. Please see our response [[here]](https://openreview.net/forum?id=LIV7-_7pYPl&noteId=l9JVRzjY_7).

---

> ### Author Response · Authors · 2022-11-17
> **Response to Reviewer whUB (1/2)**
>
> Thank you for the helpful and insightful feedback.
>
> > #### 1. Original question: "What is the design choice behind using a non-dynamic skill abstraction network? I think a temporal variant with a learnable prior would be more useful and allow for sampling of the next latent sample z."
> > #### After clarification: "I was just curious if the authors had considered a probabilistic setting. More specifically, a learned prior instead of the standard Gaussian one."
>
> We employed the standard Gaussian prior for simplicity. We agree with the reviewer and think that a learned prior would be useful in regularizing the learning process for our skill encoder, and allows for sampling the next latent sample $z$.
>
> We have followed your suggestion and conducted additional experiments where a prior network $f_{p}(z_t |h_t, a_t, ..., h_{t-K}, a_{t-K})$ is learned jointly with the posterior network $f_{q}$ (i.e., the skill encoder) using KL divergence. Concretely, we implemented the prior network using an LSTM architecture. During our iterative planning (Algo. 2), we **maintain a 10-step history of past state embeddings and actions** from executing $M$ skill latents in the previous planning iteration. For the current planning iteration, we **use the learned prior to sample the batch of future skill latents** for initializing the optimization in Eq. (1). For the initial planning iteration, we use the standard Gaussian prior to sample the skill latents for optimization.
>
> The experiment results are included below and included in Appendix H. We report the averaged normalized improvements of each approach.
>
> | Env      | Folding       |   Rope   | Bun     |  Dumpling | Wrap | Flip |
> | ---------| ------------- | -------- | ------  | --------- | ---- | ---- |
> | Ours w/o learned prior |  **0.970**   |  **0.972** | 0.874 | **0.888**  | 0.845 | **0.842** |
> | Ours w/ learned prior |  0.933   |  0.739     |  **0.907** | 0.814  | **0.881**  | 0.773 |
>
> We observed that the two approaches arrive at similar performances. We think it is an intriguing future direction to incorporate a temporal skill prior and consider other forms of exploration for tackling deformable object manipulation.
>
>
> ---
>
> > #### 2. Clarification of the prior of the skill abstraction network
>
> Thank you for your suggestion! Yes, the latent skill samples in Eq.1 are randomly initialized from the standard Gaussian prior. We have added clarification in Appendix C.2.

---

> > ### Author Response · Authors · 2022-11-17
> > **Response to Reviewer whUB (2/2)**
> >
> > > #### 3 & 4. "I think the evaluations could be fairer. The baseline models take the input from the scene in point cloud representation. A goal-conditioned policy could be well suited for the tasks introduced in this paper. I think stronger baselines would improve the quality of the submission."
> >
> > #### **[Additional baseline experiments using latent scene embedding]**
> >
> > We agree with the reviewer that the suggested evaluations could further strengthen our work. We have conducted experiments where the baseline models use the same latent embeddings as our proposed methods. The experiment results are included below and in Appendix G.
> >
> > We report the averaged normalized improvements of each approach. For convenience, we have included the original baselines and our approach.
> >
> >
> >
> > | Env      | Folding       |   Rope   | Bun     |  Dumpling | Wrap | Flip |
> > | ---------| ------------- | -------- | ------  | --------- | ---- | ---- |
> > | PPO w/o latent |  0.361  |  0.460 | 0.069 |  0.000 | 0.000 | 0.223 |
> > | PPO w/ latent |  0.630   |  0.531 | 0.000 | 0.100 | 0.082 | 0.270 |
> > | DAPG w/o latent |  0.538   |  0.246 | 0.460 | 0.000 | 0.000 | 0.000 |
> > | DAPG w/ latent |  0.459   |  0.544 | 0.294 | 0.272  | 0.183 | 0.000 |
> > | BC w/o latent |  0.685  |  0.557 | 0.379  | 0.506 | 0.134 | 0.253 |
> > | BC w/ latent |  0.672   |  0.524 | 0.000 | 0.145  | 0.433 | 0.326 |
> > | DexDeform (Ours) |  **0.970**   |  **0.972** | **0.874** | **0.888**  | **0.845** | **0.842** |
> >
> > In general, we find that the inclusion of latent scene embeddings **did not significantly improve the performance of the baseline methods**. We speculate that **the policy characteristics of the baselines are more responsible for the performance** than the designs of their perception modules.
> >
> > Specifically,
> > - For the RL approaches (PPO & DAPG), **the sample complexity becomes a limitation** under dexterous deformable object manipulation due to the large state space.
> > - Behavior cloning struggles to generalize across different goals. Because BC lacks **the ability to compositionally represent and reason about the deformation process**, which is **an inherent advantage of skill-based planning** used by our method.
> >
> > **We hope that our benchmark can inspire future research** to improve RL approaches for dexterous deformable object manipulation and overcome challenges in sample complexity (e.g., by integrating differentiable physics into RL).
> >
> >
> > #### **[Original baselines are goal-conditioned]**
> > We would like to mention that the original baseline methods are goal-conditioned policies, including BC, PPO, and DAPG. Concretely, suppose that $W^{scene}_t$ is the point cloud observation of the scene at timestep $t$, and $W^{goal}$ is the point cloud of the goal shape. The input to these baselines at timestep $t$ is the *goal-conditioned* point cloud $W_t = W^{scene}_t \cup W^{goal}$. Each point in the concatenated point cloud has a binary label that indicates whether the point belongs to the goal.
> >
> > We think that **there potentially exist other suitable approaches for learning goal-conditioned policies** for manipulating soft bodies, which is an interesting future direction to explore.
> >
> >
> > ---
> >
> > **We hope that our response has addressed your concerns, and would really appreciate it if you could raise your rating.** *Please do not hesitate to contact us if there are other clarifications or experiments we can offer.*
> >
> > Best, Authors

---

> ### Author Response · Authors · 2022-11-23
> **Looking forward to your reply**
>
> Dear Reviewer whUB,
>
> Happy Thanksgiving!
>
> Thank you again for your time. As the deadline for discussion is approaching, we hope to have a further conversation with you to see if our response resolves your concerns. We are happy to provide any additional clarifications that you may need.
>
> We have followed your suggestions to include additional experiments for investigating the learned prior. We also conducted additional baseline experiments, where the baseline models used the same latent embeddings as our proposed methods.
>
> We would appreciate you kindly checking our response. Please do not hesitate to contact us if there are further clarifications or experiments we can offer. Thanks!
>
> Best wishes, Authors

---

> > ### Comment · Reviewer_whUB · 2022-11-23
> > **Thanks for the inputs**
> >
> > I thank the authors for their extensive rebuttal. I read other reviews and the provided clarifications again. Additional experiments (1) using the latent scene representations in the baselines, (2) introducing a learned prior, and (3) using imperfect point cloud inputs to evaluate generalization performance provide more insights. I appreciate the effort. I must also say that I find it a bit surprising that more structured latent scene representations do not achieve consistent improvements.
> >
> > The rebuttal addresses all my concerns, and I do not have any further questions. I think this is an interesting paper for the community. I am in favor of accepting this paper.

---

> > > ### Author Response · Authors · 2022-11-23
> > > **Thank you**
> > >
> > > Thank you so much for your constructive and insightful feedback. We are glad to see that you appreciate our response and recognize the value of our paper for the community. Thank you again for your time.
> > >
> > > Best, Authors

---

### Official Review · Reviewer_uqMy · 2022-10-25

**Confidence:** 4
**Correctness:** 4
**Technical Novelty And Significance:** 3
**Empirical Novelty And Significance:** 3
**Recommendation:** 8

**Clarity, Quality, Novelty And Reproducibility:**

As I summarized in the above section, this paper is well-written and clear. Conceptually it is not novel but it contains good engineering to build a system. I’m also concerned about the reproducibility because if the author does not promise to release their code.

**Details Of Ethics Concerns:**

Not Applicable.

**Strength And Weaknesses:**

Strength:
1) This paper is (to the best of my knowledge) the first work studies how to do deformable object manipulation with a multi-fingered hand. Previous methods either do deformable object manipulation using parallel jaw grippers or rigid object manipulation with multi-fingered hand.
2) The writing is pretty clear and complete. It’s easy to read and understand the main message of the paper. The figures are also nice and well-designed.

Weakness:
1) Since I consider this paper as an engineering effort to build a practical system, I’m worried about if the results can be easily transferred to the real world. Specifically, this paper assumes perfect point clouds both for input and goals. However, it is unlikely to obtain perfect point cloud observations in the real world due to sensor noise. It is also hard for the user to specify point clouds as the end state.
2) The authors does not promise code release. I think realeasing code upon publication can greatly improve the paper’s impact.

**Summary Of The Paper:**

This paper propose a framework for deformable object manipulation with a dexterous multi-finger hand. It consider six tasks in simulation. The proposed framework solves these tasks by first collecting a few demonstrations by teleoperation, then it trains a skill dynamics predictor and action decoder for planning towards a given goal. The skill abstraction is defined by a fixed temporal interval. Finally, the planned action sequence is refined by gradient descent since the simulator is differentiable. The authors demonstrate success in the six tasks and it outputperform several alternatives such as reinforcement learning and imitation learning.

**Summary Of The Review:**

In summary, this paper is worth reading for a large part of the community. Though it focuses more on a system design and implementation, it can still benefit the community a lot.

----
Score Increase from 6 to 8 after discussion.

---

> ### Author Response · Authors · 2022-11-17
> **Response to Reviewer uqMy**
>
> Thank you for your constructive feedback and detailed comments.
>
> > #### 1. "I consider this paper as an engineering effort to build a practical system. I’m worried about if the results can be easily transferred to the real world. It is unlikely to obtain perfect point cloud observations in the real world. It is also hard for the user to specify point clouds as the end state."
>
>
> #### **[Novelty in the proposed method]**
>
> - We thank the reviewer for recognizing our efforts on the engineering side. We would like to highlight that the proposed method contains many innovations, such as **integrating implicit scene representation into skill learning** and **refining the learned skills with differentiable physics**.
> - As will be reported below, we find in additional experiments that our novel integration of implicit scene representation **allows for generalization across partial point clouds** with **a varying number of points observed in each frame** through four RGBD camera viewpoints.
> - Given a novel deformable shape, by integrating gradient-based trajectory optimization into the loop of skill learning, our approach expands the demonstration set and allows for generalization across novel goals.
>
> #### **[Dealing with partial point cloud input]**
>
> To further strengthen our work, we have conducted experiments taking partial point clouds observed through four RGBD camera viewpoints as inputs. The experiment results are included below and in Appendix F. We report the averaged normalized improvements.
>
> | Env      | Folding       |   Rope   | Bun     |  Dumpling | Wrap | Flip |
> | ---------| ------------- | -------- | ------  | --------- | ---- | ---- |
> | Ours w/ Perfect point cloud |  **0.970**   |  **0.972** | **0.874** | **0.888**  | **0.845** | **0.842** |
> | Ours w/ Partial point cloud |  0.954   |  0.961 | 0.814 | 0.785  | 0.809 | 0.683 |
>
> In general, we find that our use of implicit scene representation **allows for generalization across partial point clouds** with **a varying number of points observed in each frame**.
>
> While we think that it is a **crucial future direction to build perception modules that tackle other challenges** (e.g., using just a single-view camera, inferring physical properties, tackling hand occlusions), we would like to point out that our work offers a **nearly full-stack solution** to the challenging problem - learning dexterous manipulation of deformable objects. Our work provides **a simulation platform** that is fully differentiable with **a teleoperation system**, and **a novel framework** to learn perception and actions that generalize across partial point clouds.
>
> #### **[User specification of the end state ]**
>
> Many options exist for users to specify point cloud as the end state.
> - One could **use the existing shape dataset** such as ShapeNet [1] **to select the target shape**.
> - One could **use multi-view stereo inputs to extract point-cloud representations of shapes** in the real world.
> - One could also **use 3D computer graphics software** such as Blender **to modify the existing shapes or create new shapes**.
>
> All three approaches mentioned above could provide point cloud representations of different shapes for the user to specify the end state. It is also feasible to create **simple API to import shapes** or **interactive VR applications to draw shapes** for users.
>
> [1] Chang, Angel X., et al. "Shapenet: An information-rich 3d model repository." arXiv preprint arXiv:1512.03012 (2015).
>
> ---
>
> > #### 2. "The authors do not promise code release. I think releasing code upon publication can greatly improve the paper’s impact."
>
> Thank you so much for your suggestion! We promise to release the code of our entire pipeline, containing a differentiable simulation platform, a human teleoperation system, and our learning-based framework. We also promise to release our human demonstration dataset.
>
> ---
>
> **We hope that our response has addressed your concerns, and would really appreciate it if you could raise your rating.** *Please do not hesitate to contact us if there are other clarifications or experiments we can offer.*
>
> Best, Authors

---

> > ### Comment · Reviewer_uqMy · 2022-11-21
> > **Response**
> >
> > I thank the authors for their detailed response to my review. The authors well-addressed my questions, especially about the possiblity for sim2real transfer and about the open-source plan. Therefore I decide to increase my score.

---

> > > ### Author Response · Authors · 2022-11-23
> > > **Thank you**
> > >
> > > Thank you for your detailed and constructive comments. We are glad to see that you appreciate our response. Thank you again for your time.
> > >
> > > Best, Authors

---

### Author Response · Authors · 2022-11-17
**General Response to All Reviewers**

We would like to thank the reviewers for their thoughtful and constructive feedback.

We are glad to see that reviewers generally appreciated our paper: the first investigation on dexterous deformable object manipulation (uqMy), the usefulness of our differentiable simulation and teleoperation platform to the community (uqMy), leveraging implicit scene representation to overcome challenges of modeling soft bodies  (whUB), an interesting set of simulated benchmarking tasks (WWJM), and writing clarity (uqMy, whUB)

We would like to emphasize again that our primary contributions are
- We perform, to the best of our knowledge, the first investigation on the learning-based dexterous manipulation of deformable objects.
- We build a platform that integrates a low-cost teleoperation system with a soft-body simulation that is differentiable, allowing humans to provide demonstration data.
- We propose DexDeform, a principled framework that abstracts dexterous manipulation skills from human demonstration, and refines the learned skills with differentiable physics.
- Our approach outperforms the baselines and successfully accomplishes six challenging tasks such as Flip, learning complex soft-body manipulation skills from demonstrations.

According to reviewers' comments, we have provided a detailed response to each reviewer's questions and concerns. We have revised our paper to include the following changes:

- We have added experiments on using partial point clouds observed through four RGBD camera viewpoints as inputs in Appendix F (reviewer uqMy).
- We have clarified that the latent skill samples in Eq.1 are randomly initialized from the standard Gaussian prior in Appendix C.2 (reviewer whUB).
-  We have added experiments where our proposed method is trained with a learned prior in Appendix H (reviewer whUB).
-  We have added experiments where the baseline models use the same latent embeddings as our proposed methods in Appendix G (reviewer whUB).
- We have brought Algorithm 2 (now Algo. 1) over to Sec. 2.3, reorganized the structure of Sec. 2.3, and standardized our notation throughout the paper. (reviewer WWJM).
- We have provided a new figure (Fig. 6) in Appendix D that illustrates the training and inference processes of our proposed method (reviewer WWJM).
- We have added details of our environments in Appendix A (reviewer WWJM).


We promise to release the code of our entire pipeline, containing a differentiable simulation platform, a human teleoperation system, and our learning-based framework. We also promise to release our human demonstration dataset.



We hope our responses have convincingly addressed all reviewers’ concerns. We thank all reviewers’ time and efforts again! Please don’t hesitate to let us know of any additional comments on the manuscript or the changes.

---

### Author Response · Authors · 2022-12-12
**Summary of our rebuttal and discussion**

We sincerely thank all reviewers and ACs for their efforts and time in reviewing our paper and their constructive suggestions that strengthen our work. We deeply appreciate the positive 8-8-6 evaluation from the reviewers.

To summarize our response:

#### [Contribution]

We would like to emphasize again that our primary contributions are
- We perform, to the best of our knowledge, the first investigation on the learning-based dexterous manipulation of deformable objects.
- We build a platform that integrates a low-cost teleoperation system with a soft-body simulation that is differentiable, allowing humans to provide demonstration data.
- We propose DexDeform, a principled framework that abstracts dexterous manipulation skills from human demonstration, and refines the learned skills with differentiable physics.
- Our approach outperforms the baselines and successfully accomplishes six challenging tasks such as Flip, learning complex soft-body manipulation skills from demonstrations.

#### [Additional Experiments]
- We have added experiments on using partial point clouds observed through four RGBD camera viewpoints as inputs in Appendix F (reviewer uqMy).
- We have added experiments where our proposed method is trained with a learned prior in Appendix H (reviewer whUB).
- We have added experiments where the baseline models use the same latent embeddings as our proposed methods in Appendix G (reviewer whUB).

#### [Writing]
- We have clarified that the latent skill samples in Eq.1 are randomly initialized from the standard Gaussian prior in Appendix C.2 (reviewer whUB).
- We have brought Algorithm 2 (now Algo. 1) over to Sec. 2.3, reorganized the structure of Sec. 2.3, and standardized our notation throughout the paper. (reviewer WWJM).
- We have provided a new figure (Fig. 6) in Appendix D that illustrates the training and inference processes of our proposed method (reviewer WWJM).
- We have added details of our environments in Appendix A (reviewer WWJM).

#### [Code Release]

- We promise to release the code of our entire pipeline, containing a differentiable simulation platform, a human teleoperation system, and our learning-based framework. We also promise to release our human demonstration dataset.

We owe many thanks to the reviewers for their insightful suggestions, which help improve our paper a lot. The additional experiments and writing modifications will be reflected in the final version of our paper as well.

Best, Authors

---

### Decision · Program_Chairs · 2023-01-20

**Decision:**

Accept: poster

**Justification For Why Not Higher Score:**

Overall the paper seems to try to do a bit too much for the given space-limit. E.g. the designed teleportation setup only seems tangentially realted to the rest. The paper focuses on a specific setting, while the ideas and methods are potentially more broadly applicable. It remains a bit unclear what task specific and what more generally applicable.

**Justification For Why Not Lower Score:**

All reviewers argue for acceptance.

**Metareview: Summary, Strengths And Weaknesses:**

Summary:
The paper studies the manipulation of deform-able objects with a dexterous hand. They combine human demonstrations, with differentiable physics guided exploration, abstraction, and refinement. The approach is evaluated experimentally in 6 tasks

Strengths:
- Novel application domain
- Complete framework
- Works well experimentally

Weaknesses:
- Real-world experiments: the proof is in the pudding - especially since this setting is very hard to simulated realistically (de-formable objects / contacts)
- Quite a complex system with many moving components

**Note From Pc:**

if the above contains the word "oral" or "spotlight" please see: "oral" presentation means -> notable-top-5% and "spotlight" means -> notable-top-25%. As stated in our emails, we are disassociating presentation type from AC recommendations

**Summary Of Ac-Reviewer Meeting:**

N/A